# Effect of Blood Flow Restriction during the Rest Periods of Squats on Accuracy of Estimated Repetitions to Failure

**DOI:** 10.3390/sports12010014

**Published:** 2023-12-31

**Authors:** Daniel A. Hackett, Boliang Wang, Derek L. Tran

**Affiliations:** 1Discipline of Exercise and Sports Science, Sydney School of Health Sciences, Faculty of Medicine and Health, The University of Sydney, Sydney 2006, Australia; bwan9245@uni.sydney.edu.au (B.W.); derek.tran@sydney.edu.au (D.L.T.); 2Central Clinical School, The University of Sydney School of Medicine, Sydney 2006, Australia; 3Department of Cardiology, Royal Prince Alfred Hospital, Sydney 2006, Australia

**Keywords:** resistance training, repetition maximum, training intensity, fatigue, weightlifting

## Abstract

This study investigated the impact of resistance training with blood flow restriction during rest (BFR_rest_) on the accuracy of estimated repetitions to failure (ERF). It also explored associations between error in ERF and mean concentric velocity (MCV) along with physiological responses. In a randomised cross-over study, 18 male trainers (23.4 ± 2.7 years) performed three sets of squats at 70% of their one-repetition maximum until failure. One session integrated BFR_rest_, while another employed traditional passive inter-set rest (TRAD) during the 3 min inter-set rest intervals. Cardiorespiratory and metabolic measures were taken in the inter-set recovery periods. The results revealed no significant differences between BFR_rest_ and TRAD in terms of ERF and error in ERF. A notable set effect for ERF was observed, with a greater ERF during set 1 compared to sets 2 and 3 (*p* < 0.001). Additionally, a lower error in ERF was observed during sets 2 and 3 compared to set 1 (*p* < 0.001). Error in ERF were strongly associated with the respiratory exchange ratio, and moderately associated with end-tidal carbon dioxide partial pressure, carbon dioxide output, and MCV variables. Notably, the precision of ERF seems to be predominantly influenced by indicators of physiological stress rather than the incorporation of BFR_rest_.

## 1. Introduction

The integration of blood flow restriction (BFR) during inter-set rest intervals, referred to as BFR_rest_, in the context of high-load resistance training (HL-RT), is a practice believed to disrupt the recovery process, impacting oxygenation and metabolite clearance after a set of repetitions [1]. This disruption potentially leads to an accumulation of metabolites before the commencement of the subsequent set, a phenomenon recognised as metabolic freeze [2]. Despite its potential implications, there is limited research exploring the acute effects of HL-RT when performed with BFR_rest_ in comparison to traditional inter-set passive rest [3,4]. Notably, one study found that BFR_rest_ intensifies metabolic stress [4], while another study indicated that this practice hinders recovery processes [3].

The intricate dynamics of techniques employed to manipulate inter-set recovery processes, such as BFR_rest_, raise questions about their influence on the lifter’s ability to estimate repetitions to failure (ERF). The ERF scale, also known as repetitions in reserve (RIR), has garnered considerable attention in monitoring resistance training intensity over the past five years [5,6,7,8,9,10,11]. The accuracy of ERF hinges on the proximity to concentric failure, with greater accuracy achieved as a lifter approaches concentric failure [6,10]. However, the current body of knowledge lacks insights into how resistance training, particularly when combined with BFR (either during muscle contractions or inter-set rest), may influence the accuracy of ERF during a set to failure. This gap in understanding underscores the need for further investigation into the intricate interplay between BFR, inter-set recovery strategies, and the lifter’s capacity to gauge repetitions to failure accurately.

Research has demonstrated that individuals with >1 year of resistance training experience exhibit a higher level of accuracy in ERF compared to those with <1 year of experience [11]. This heightened accuracy in experienced trainers may be attributed to their increased familiarity with exertional sensations and the corresponding mechanical performance associated with prolonged resistance training engagement [12]. As lifters accumulate more experience in resistance training, it is plausible that they develop a nuanced understanding of exertional sensations and the intricacies of mechanical performance during exercise. When performing sets of a resistance exercise, as a lifter approaches concentric failure, there is a well-documented pattern of escalating exertion levels alongside a decrease in barbell mean concentric velocity (MCV) [8,9]. This suggests a dynamic interplay between the lifter’s perceived exertion and the observable changes in mechanical performance. Both these factors may contribute significantly to the enhanced accuracy of ERF, as the lifter gains experience and becomes attuned to the nuanced signals that accompany nearing concentric failure.

Despite the acknowledgment of these important connections between exertional sensations, mechanical performance, and ERF accuracy, there remains a noticeable gap in research. Specifically, there is insufficient exploration into whether alterations in cardiorespiratory and metabolic responses during a resistance training session exert an influence on ERF and, consequently, the precision of ERF. This underlines the need for further investigations to unravel the complex relationships between physiological responses, the lifter’s subjective experience, and the accuracy of ERF in resistance training scenarios.

Since BFR_rest_ is a novel BFR practice that may be used by resistance trainers to promote a greater exercise stimulus, it is of interest to investigate whether heightened physiological responses influence ERF accuracy and, thus, impact monitoring exercise performance. Therefore, the primary objective of this study was to examine the impact of HL-RT performed with BFR_rest_ in comparison to traditional inter-set passive rest (TRAD) on ERF and the accuracy of ERF. A secondary aim was to explore the potential associations between barbell MCV and various physiological responses, encompassing cardiorespiratory and metabolic factors, subsequent to sets of resistance exercises, with ERF and the accuracy of ERF. Our hypothesis posited that no discernible differences would manifest between BFR_rest_ and TRAD concerning ERF and the accuracy of ERF. Furthermore, we anticipated that MCV and a myriad of physiological responses would exhibit associations with ERF and the accuracy of ERF. The insights generated from this study hold potential significance for a greater understanding of the multifaceted factors influencing ERF when employed as a monitoring tool during resistance training.

## 2. Materials and Methods

### 2.1. Participants

A total of eighteen healthy young males (mean age: 23.4 ± 2.7 years, height: 177.8 ± 5.9 cm, body mass: 84.7 ± 7.9 kg) participated in this study. The inclusion criteria required participants to be male, aged between 18 and 45 years, possess a minimum of six months of resistance training experience (including regular squats), and maintain overall health (free from musculoskeletal conditions and chronic diseases). All potential participants were provided with comprehensive information about the study’s purpose, associated risks, benefits, and experimental procedures. Before initiating the study, each participant provided written informed consent. The study adhered to the approved protocol established by the University of Sydney Human Research Ethics Committee (Project No.: 2019/603)

Each participant made three visits to the university exercise laboratory. The initial visit involved establishing their one-repetition maximum (1RM) for the barbell back squat and familiarising them with the equipment and procedures. The last two visits encompassed the experimental sessions. To prepare for each visit, participants were instructed to refrain from engaging in strenuous physical activity in the 24–48 h preceding the session, abstain from consuming caffeine or pre-workout supplements within 2–3 h before the session, and avoid eating within 1 h before the visit. While the participants were not obligated to adhere to a specific diet before each visit, the experimental sessions were consistently conducted at a similar time of day. The sessions were separated by ≥72 h for recovery.

### 2.2. Experimental Design

An acute crossover experimental design with a random treatment order was employed to investigate the impact of BFR_rest_ versus traditional inter-set rest on ERF. The participants underwent a high-load squat protocol to volitional fatigue with (1) BFR during the 3 min inter-set rest (BFR_rest_), and (2) traditional passive rest during the 3 min inter-set rest (i.e., without BFR_rest_) (TRAD). During the experimental sessions, the participants reported their ERF during each set after completing 5 repetitions of squats, followed by continuing to perform repetitions to volitional fatigue. Cardiorespiratory variables (respiratory gases and heart rate) and blood lactate levels were measured during the inter-set recovery periods. Additionally, the mean concentric velocity (MCV) of the barbell was recorded for each repetition.

### 2.3. Pre-Testing Session (Visit 1)

Within this visit, the 1RM assessment for the barbell back squat was performed. Prior to the 1RM test, the participants engaged in a warm-up comprising 8–10 repetitions with the barbell. This was followed by two sets of 4–6 repetitions with progressively increasing submaximal loads (e.g., 50% and 70% of estimated 1RM). In the 1RM protocol, the participants executed a singular lift with incremental load adjustments (5–10% increments) after each successful attempt. Adequate rest periods of 3–5 min were provided between attempts. The cycle persisted until the participants could no longer successfully complete a lift, determining their 1RM as the maximum weight lifted.

A successful 1RM squat attempt entailed descending through knee and hip flexion until the thighs reached a parallel position with the floor, followed by ascending to a fully upright stance. Following the 1RM test, the participants underwent orientation with the equipment and procedures for the subsequent experimental sessions. Notably, KAATSU Air Bands (Sato Sports Plaza, Tokyo, Japan) were affixed to their legs and inflated to the prescribed pressure corresponding to the BFR_rest_ condition. This served to familiarise them with the sensation before the experimental interventions.

### 2.4. Experimental Sessions (Visits 2 and 3)

Before engaging in exercise, resting cardiorespiratory and metabolic measures were gathered, a detailed explanation of which is provided below. Following this, the participants underwent a standardised warm-up, involving 10 repetitions of squats at 40% of their one-repetition maximum (1RM), followed by a 30 s rest, and then 8 repetitions at 60% 1RM. In both the BFR_rest_ and TRAD sessions, the participants performed 3 sets of barbell back squats at 70% 1RM until reaching volitional fatigue, with a 3 min rest between sets. The eccentric phase of the squat required the thigh to be parallel to the floor, and the participants were instructed on the proper depth. Sets were terminated if incorrect technique or failure to complete the concentric phase occurred. The participants focused on a controlled eccentric phase, lasting approximately 2 s, and maximal concentric velocity, with researchers providing verbal encouragement during the experimental sessions.

In both the BFR_rest_ and TRAD sessions, we employed the KAATSU Master (Sato Sports Plaza, Tokyo, Japan) with large-size cuffs worn by all participants (5 cm width and 60–70 cm length). The KAATSU cuff was wrapped around the proximal thigh below the inguinal fold of each leg. Notably, the participants wore the KAATSU cuff just before the first set of squats until the end of the rest period for the third set of squats. After completing a set of squats, the participants immediately sat down on a chair and rested for 3 min. In the BFR_rest_ session, upon sitting down to rest, the cuff was inflated to 290–320 mmHg, representing 73–80% of the device’s maximum pressure (i.e., 400 mmHg). Once the rest time was completed, the cuff was deflated. In the TRAD session, the cuff remained deflated (i.e., negligible pressure) during the rest period.

### 2.5. Estimated Repetitions to Failure

The participants received instructions on utilising the ERF scale before embarking on the initial experimental session. To facilitate the linking of exercise intensities with the complete response range of the ERF scale, the participants were guided to employ a memory-anchoring procedure [13]. This involved prompting the participants to reflect on instances during their training when their exertion levels matched estimations at both ends of the ERF scale. For instance, an estimation of “0” signified the inability to complete additional repetitions (reaching momentary failure), while “10” indicated the potential to perform 10 or more repetitions. When presenting the ERF scale to participants, the precise wording used was “how many additional repetitions can you perform?” The ERF scale was affixed to a wall at eye level when standing, positioned approximately 1 m in front of where the participants executed their squats. The participants were directed to report their ERF after the concentric phase of the fifth repetition in each set, which involved a brief pause followed by continuing the set until failure.

### 2.6. Cardiorespiratory and Metabolic Measures

Respiratory gases were continuously monitored through open-circuit spirometry while in a seated position (COSMED Indirect Calorimetry system, Rome, Italy; Ultima Series CardiO2 and PFX, Medgraphics, Minneapolis, MN, USA). Simultaneously, heart rates were recorded using the Polar T31 (Polar Electro Oy, Kempele, Finland). Initial measurements were taken before the commencement of exercise (representing resting values), and subsequent readings were captured after each set of squats. The analysis primarily focused on the recovery phase due to more pronounced responses in comparison to the exercise phase during resistance training [14].

The measured parameters included the absolute oxygen uptake (VO_2_), minute ventilation (VE), carbon dioxide output (VCO_2_), respiratory exchange ratio (RER), tidal volume, breath rate, end-tidal oxygen partial pressure (PETO_2_), and end-tidal carbon dioxide partial pressure (PETCO_2_). Furthermore, heart rate readings were logged during the recovery periods between sets. Mean values derived from each 3 min interval were utilised for subsequent analysis. To assess capillary blood lactate levels, samples were collected in the final 30 s of the inter-set rest using a portable analyser (Lactate Scout 4, EΚF Diagnostics, Cardiff, UK) from either the finger or earlobe, with the sampling site consistent for each participant.

### 2.7. Mean Concentric Velocity

The barbell MCV was evaluated in all sets utilising the GymAware linear position transducer (Kinetic Performance Technology, Mitchell, Australia). The MCV was determined by dividing the barbell displacement by the duration of the concentric phase (from the start of vertical movement to the end). The reported outcomes included the MCV during a set (MCV), MCV loss calculated as 100 × (MCV_last_ − MCV_best_)/MCV_best_), and MCV loss (5th repetition) calculated as 100 × (MCV_5th repetition_ − MCV_best_)/MCV_best_) [15].

### 2.8. Statistical Analyses

The data were presented as means ± SD and analysed using SPSS version 28.0 for Windows (IBM Corp., Armonk, NY, USA). Data normality was assessed via the Shapiro–Wilk test, considering skewness and kurtosis measures within acceptable ranges (kurtosis < +2.0, >−2.0; skewness between +2.0 and −2.0) [16]. The accuracy of the ERF was determined by the error in ERF, which is defined as the difference between the ERF and the actual repetitions to failure for a set. The error in ERF was calculated in two ways. The first method involved the absolute difference, where the sign of the numbers was ignored, meaning that negative numbers were converted to positive. This was referred to as the error in ERF (absolute). The second method involved recognising the sign of numbers, thereby acknowledging underestimation (negative values) and overestimation (positive values) of the ERF. This was referred to as the error in ERF (signed).

A two-way analysis of variance (ANOVA) was employed to assess the effects of BFR_rest_ versus TRAD rest on ERF. Post hoc tests, corrected using Bonferroni, were conducted for significant interaction effects to identify differences between sets and condition × sets. Effect sizes were quantified using the partial eta-squared (η^2^) value [17]. The effect sizes were categorised as small (η^2^ = 0.01 to <0.06), medium (η^2^ = 0.06 to <0.14), or large (η^2^ ≥ 0.14) [18].

Relationships between response variables (VE, VO_2_, VCO_2_, RER, breath rate, tidal volume, PETO_2_, PETCO_2_, blood lactate, heart rate, MCV, MCV loss, and MCV loss (5th repetition)), ERF, and error in ERF (absolute and signed) were assessed using the Pearson correlation coefficient (*r*). The strength of the correlations was qualitatively assessed as trivial (*r <* 0.1), small (*r* > 0.1–0.3), moderate (*r* > 0.3–0.5), strong (*r* > 0.5–0.7), very strong (*r* > 0.7–0.9), nearly perfect (*r* > 0.9), or perfect (*r* = 1.0) [19]. Statistical significance was set at *p* < 0.05 for all analyses.

## 3. Results

### 3.1. Characteristics of Participants

Table 1 displays the muscle strength and resistance training background of the participants. The participants were classified as having an advanced resistance training status due to squatting >150% of their body weight and having consistent resistance training experience for several years (although interruptions were not determined), with a good frequency of approximately four sessions per week [20].

### 3.2. Effect of BFR_rest_ versus TRAD on ERF

There was no significant difference between BFR_rest_ and TRAD for cardiorespiratory and metabolic responses at rest. However, among the cardiorespiratory and metabolic responses following sets of squats, a significant difference was observed solely in PETCO_2_ between the two conditions. Specifically, lower values were evident for BFR_rest_ (33.94 mmHg) compared to TRAD (35.35 mmHg), with a *p*-value of 0.026 and a partial η² of 0.049, indicating a potential greater metabolic stress associated with the former condition.

No significant condition effect or condition–set interaction was observed for the ERF and error in ERF (absolute and signed). However, a significant set effect was identified for the ERF (*p* < 0.001, partial η^2^ = 0.29). A post hoc analysis revealed that the ERF was significantly greater for set 1 (6.0 ± 2.0) compared to both set 2 (4.4 ± 1.7, *p* < 0.001) and set 3 (3.4 ± 1.3, *p* < 0.001) (Figure 1).

Furthermore, a significant set effect for error in ERF (signed) was observed (*p* < 0.001, partial η^2^ = 0.61), as well as for error in ERF (absolute) (*p* < 0.001, partial η^2^ = 0.44). Post hoc testing indicated that the error in ERF (signed) was significantly greater for set 1 (5.5 ± 3.0) compared to set 2 (1.3 ± 2.2, *p* < 0.001) and set 3 (0.5 ± 1.9, *p* < 0.001) (see Figure 2). Similarly, the post hoc analysis of error in ERF (absolute) revealed that set 1 (5.5 ± 3.0) was less accurate compared to set 2 (1.9 ± 1.6, *p* < 0.001) and set 3 (1.3 ± 1.4, *p* < 0.001).

### 3.3. Correlations between Resistance Exercise Response Variables and ERF

ERF exhibited a strong association with VCO_2_ (*r* = 0.531, *p* < 0.001), and moderate associations with RER (*r* = 0.441, *p* < 0.001) and PETCO_2_ (*r* = 0.456, *p* < 0.001). Additionally, moderate to small associations for ERF were observed with tidal volume, blood lactate, MCV, MCV loss, and MCV loss (fifth repetition) (*r* = −0.252 to 0.335, *p* < 0.05) (Table 2).

The error in ERF (signed) demonstrated a strong association with RER (*r* = 0.641, *p* < 0.001), and moderate associations with PETCO_2_ (*r* = 0.488, *p* < 0.001), VCO_2_ (*r* = 0.491, *p* < 0.001), and MCV loss (*r* = −0.385, *p* < 0.001) (Figure 3A–D). Moderate to small associations for error in ERF (signed) were also found with breath rate, tidal volume, blood lactate, MCV, and MCV loss (fifth repetition) (*r* = −0.276 to 0.349, *p* < 0.05) (Table 2). Concerning the error in ERF (absolute), a strong association was observed with RER (*r* = 0.638, *p* < 0.001) and VCO_2_ (*r* = 0.526, *p* < 0.001), along with a moderate association with PETCO_2_ (*r* = 0.486, *p* < 0.001). Furthermore, moderate to small associations for error in ERF (absolute) were found with breath rate, tidal volume, blood lactate, MCV, MCV loss, and MCV loss (fifth repetition) (*r* = −0.252 to 0.335, *p* < 0.05) (Table 2).

## 4. Discussion

This exploratory study examined the impact of incorporating BFR_rest_ in high-load resistance training on ERF accuracy compared to traditional passive inter-set rest. The results aligned with the hypothesis, showing no influence of BFR_rest_ on ERF or error in ERF. Although the first set exhibited higher ERF and less accuracy, no differences were noted between the second and third sets. The study also explored whether ERF and ERF accuracy was associated with physiological responses and barbell MCV. Strong to moderate correlations were found for VCO_2_, RER, and PETCO_2_ with ERF and error in ERF. Additionally, moderate to small correlations were found for MCV and MCV loss with ERF and error in ERF. Therefore, the accuracy of ERF seems to be predominantly influenced by indicators of physiological stress rather than the incorporation of BFR_rest_. These insights further the understanding on the complex factors influencing ERF as a monitoring tool in resistance training.

The utilisation of BFR_rest_ in exercise is purported to disrupt the recovery process when compared to conventional resistance training practices [1]. However, the findings from the present study did not consistently demonstrate a significant influence of BFR_rest_ on impairing recovery, with only PETCO_2_ exhibiting differences between conditions. A decrease in PETCO_2_ is commonly observed post high-intensity exercise, indicating a hyperventilation response [21]. Enhanced ventilation is required during the recovery period of high-intensity exercise to improve metabolic acidosis, usually accompanied by greater VCO_2_ production. The lower PETCO_2_ for BFR_rest_ compared to TRAD suggests some evidence that BFR_rest_ may impede recovery. Nevertheless, the primary factor influencing both ERF and error in ERF was identified as the physiological stress incurred from repeating sets to volitional fatigue, irrespective of the condition (BFR_rest_ versus TRAD). Notably, any potential discomfort or sensation experienced from inflating the cuffs during the recovery period did not seem to affect the participants’ error in ERF.

Previous research has indicated that the error in ERF decreases when there is a reduction in the true proximity to concentric failure [7]. However, minimal differences in the accuracy of ERF have been observed when reporting ERF with a lower number of repetitions to concentric failure (i.e., <5 repetitions) [6]. This may explain the similar error in ERF between sets 2 and 3 in the present study. The initial set would have induced metabolic stress that led to reduced performance (i.e., a lower number of repetitions) and likely heightened exertional sensations (e.g., muscle activation, afferent signals from Golgi tendon organs, muscle spindles, and mechanoreceptors) in the subsequent sets. Interestingly, the participants tended to underestimate their repetitions to failure in the initial sets, while in the later sets, there was a tendency for some participants to overestimate their repetitions to failure.

The associations observed between various respiratory variables related to the production of CO_2_ and ERF along with the error in ERF underscore the significant impact of anaerobic metabolism on the reporting and accuracy of ERF. Following a resistance exercise set, VO_2_ decreases towards resting values, while CO_2_ production rapidly increases—a characteristic of a system in a non-steady state [14]. The present study represents the first known investigation to date that delineates the influence of metabolic stress on ERF and its error, drawing attention to robust-to-moderate associations with variables such as VCO_2_, RER, and PETCO_2_. While it is imperative to acknowledge that correlation does not imply causation, the evidence suggests that metabolic stress indirectly contributes to the accuracy of ERF. As the volume of resistance training increases, the respiratory system faces a substantial demand due to heightened metabolic stress. This was demonstrated by Hackett et al. [22], where performing five sets versus two sets across various resistance exercises resulted in an increased hyperventilatory response, evident through decreases in PETCO_2_. Notably, all sets in our study were executed to volitional fatigue, with the concentric phase performed at maximal concentric velocity. This implies that neuromuscular fatigue was likely more pronounced in our study compared to scenarios where sets were not performed to failure and a self-selected repetition cadence was employed [23]. Therefore, factors such as training volume, sets to failure, and repetition cadence emerge as crucial elements that are likely to influence metabolic stress and, consequently, impact the accuracy of ERF. Recognising these interconnections enhances our understanding of the multifaceted relationship between respiratory variables, metabolic stress, and the intricate dynamics of ERF.

The relationship between squat performance and ERF was explored in this study, revealing noteworthy associations, particularly with MCV and MCV loss. The monitoring of MCV is a well-established practice in resistance exercise, serving as a valuable metric for assessing training stress and recovery in order to optimise physiological adaptations [24]. In the context of traditional resistance training sets, MCV typically exhibits a decline as repetitions approach concentric failure [9]. This decline stems from transient reductions in muscle fibre shortening speeds, relaxation times, and force production induced by fatigue [25]. Consequently, monitoring changes in MCV or observing increased MCV loss during resistance training performed with maximal voluntary effort on every repetition can serve as an indicator of fatigue [26]. In our study, the observed moderate associations between MCV variables and accuracy in ERF align with existing knowledge in the field [13]. The heightened metabolic stress evident in increased respiratory responses, coupled with declines in MCV variables, provides a more comprehensive understanding of the impact of fatigue on the accuracy of ERF.

Previous studies exploring ERF, also known as repetitions in reserve (RIR), have paid scant attention to the directional accuracy of predictions, specifically whether individuals tend to underpredict or overpredict their performance [5,6,7,8,9,10,11]. A distinctive facet of this study lies in our attempt to elucidate whether physiological or performance variables influence the direction of estimation. An examination of the coefficient of correlations for the error in ERF (signed) data indicated that the participants tended to underestimate their ERF before inducing a higher metabolic disturbance during a set (i.e., performing a greater number of repetitions). However, the effects of fatigue on squat performance became evident as the sets progressed (i.e., sets 2 and 3), leading to a reduction in the volume of repetitions and likely mitigating the metabolic stress. This phenomenon explains the observed trend towards greater accuracy with less metabolic stress and change in fatigue, as reflected by the parameter of MCV loss.

The most substantial change in accuracy appeared to occur after a major shift in physical stress and squat performance, such as in the first set performed to concentric failure. Subsequently, as the sets continued, less pronounced alterations in physical stress and performance, attributed to the presence of fatigue, seemed to contribute to more accurate ERF. Interestingly, there was a tendency for some participants to overpredict repetitions to failure. This overprediction could stem from participants striving to achieve a specific target ERF rather than accurately reflecting their true perceived abilities. Alternatively, the relative constancy in performance or exertional cues from the outset until the point of reporting ERF might have misled participants into believing they could sustain their performance for a longer duration than was actually feasible. While these insights shed light on the directional tendencies of ERF accuracy, further research is warranted to gain a deeper understanding of the factors influencing overpredicting versus underpredicting repetitions to failure. This exploration holds promise for refining exercise prescription strategies and tailoring interventions to individualised perceptions and responses to fatigue.

The interpretation of the current study’s findings necessitates an acknowledgment of several limitations. The participants involved in this research possessed an average of approximately 5 years of resistance training experience. However, crucial information regarding their typical training prescription, including details such as sets, loads, exercises, and recovery periods between sets, was not collected. The variability in training history among subjects could have potentially impacted the results, underscoring the need for consideration in future research studies exploring this topic. Moreover, the relative strength of the participants, with a squat performance of 1.7 times their body weight, falls below the reported values for similar-aged male powerlifters, who typically exhibit a ratio of approximately 2.3 times their body weight [27]. Consequently, it is plausible that the outcomes of the present study may differ within a cohort of stronger resistance trainers, emphasizing the importance of further investigations in this regard. Additionally, there is a possibility that the participants may have chosen to conclude a set based on reaching their ERF during the resistance exercises. While it remains challenging to definitively ascertain the occurrence of such decisions throughout the performances, it is noteworthy that all participants received consistent encouragement across all sets to execute as many repetitions as possible to concentric failure. This approach was implemented to minimise the risk of participants prematurely terminating sets based on reaching their ERF, although it cannot be ruled out entirely. Finally, the participants were not instructed to follow a specific diet prior to each experimental session, which may have influenced their physical performance and perceptual responses. The potential confounding factors which have been acknowledged may assist future studies with enhancing the robustness and validity of their findings.

## 5. Conclusions

Our exploratory study found that incorporating BFR_rest_ in high-load resistance training did not significantly impact ERF accuracy compared to traditional passive inter-set rest. The initial sets for both conditions displayed higher ERF and lower accuracy; however, lower ERF and greater accuracy of ERF was observed in subsequent sets. Notably, our findings indicate that the accuracy of ERF is primarily influenced by physiological stress indicators such as VCO_2_, RER, and PETCO_2_, as well as barbell MCV metrics, rather than the introduction of BFR_rest_. These insights contribute to the understanding of ERF as a monitoring tool in resistance training and provide avenues for future research.

## Figures and Tables

**Figure 1 sports-12-00014-f001:**
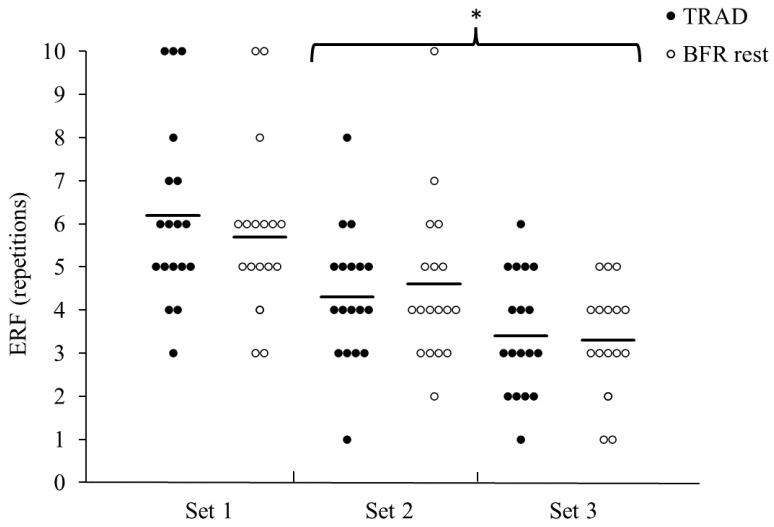
Estimated repetitions to failure (ERF) for sets 1–3 during squats for traditional inter-set rest (TRAD) and blood flow restriction during rest (BFR_rest_). Individual participant data represented by a dot for each set and condition. Solid lines represent the condition mean values. * Significantly lower ERF compared to set 1 (*p* < 0.001).

**Figure 2 sports-12-00014-f002:**
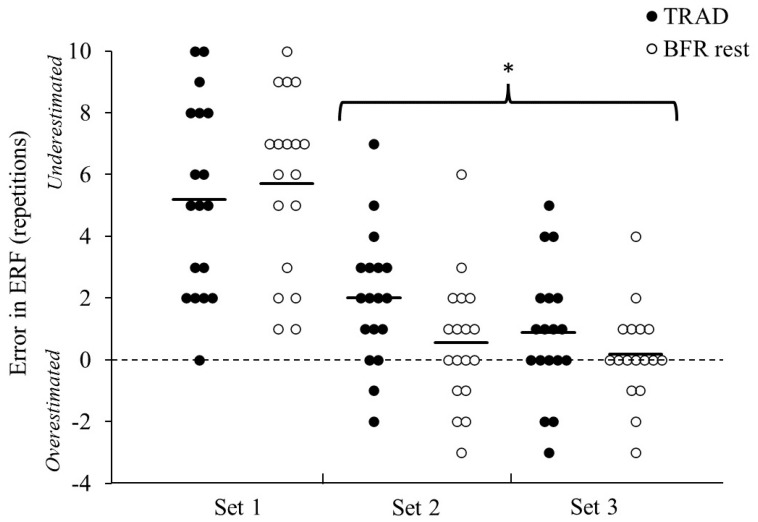
Error in estimated repetitions to failure (error in ERF) for sets 1–3 during squats for traditional inter-set rest (TRAD) and blood flow restriction during rest (BFR_rest_). Individual participant data represented by a dot for each set and condition. Solid lines represent the condition mean values. * Significantly higher error in ERF compared to set 1 (*p* < 0.001).

**Figure 3 sports-12-00014-f003:**
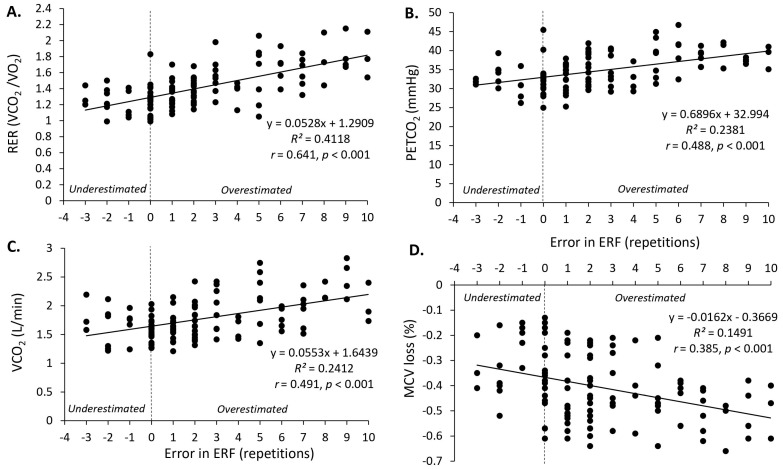
Associations between error in estimated repetitions to failure (ERF (signed)) and RER (**A**); PETCO_2_ (**B**); VCO_2_ (**C**); and MCV loss (**D**). Negative numbers on the *x* axis represent an underestimation in accuracy for ERF and positive numbers show an overestimation in the accuracy of ERF. Significant correlations for all these analyses were found at *p* < 0.001.

**Table 1 sports-12-00014-t001:** Characteristics of participants.

Muscle Strength	Results
1RM squat (kg)	141.9 ± 20.6
1RM squat (kg 1RM/kg BM)	1.7 ± 0.2
**Exercise Background**	
Years of RT	5 ± 2.4
RT sessions per week	4.2 ± 2.2
Squat sessions per week	1.5 ± 0.9

1RM = one-repetition maximum; BM = body mass; RT = resistance training.

**Table 2 sports-12-00014-t002:** Pearson correlation coefficients of cardiorespiratory, metabolic, and barbell velocity with ERF and ERF accuracy.

Variables	ERF	Error in ERF (Absolute)	Error in ERF (Signed)
VE	0.17	0.135	0.097
VO_2_	0.119	−0.125	−0.191
Breath rate	−0.125	−0.273 **	−0.29 **
Tidal volume	0.231 *	**0.309 ****	0.285 **
PETO_2_	0.019	0.135	0.118
Blood lactate	−0.252 **	**−0.30 ****	−0.276 **
Heart rate	0.062	−0.043	−0.166
MCV	0.246 *	**0.345 ****	**0.349 ****
MCV loss (5th repetition)	**0.335 ****	0.249 **	0.204 *

ERF = estimated repetitions to failure; error in ERF = difference between ERF and actual repetitions to failure; error in ERF (absolute) = negative numbers were converted to positive; error in ERF (signed) = recognising the sign of numbers; VE = minute ventilation; VO_2_ = volume of oxygen consumption; MCV = mean concentric velocity; end-tidal oxygen partial pressure (PETO_2_). Values in bold signify moderate associations. * Significant at *p* < 0.05; ** significant at *p* < 0.01.

## Data Availability

The data are available on reasonable request from the corresponding author in agreement with the institutional ethical requirements.

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
