# Peer review of "Effect of Blood Flow Restriction during the Rest Periods of Squats on Accuracy of Estimated Repetitions to Failure"

_sports, 2023, doi:10.3390/sports12010014_

Round 1

Reviewer 1 Report

Comments and Suggestions for Authors

This well written manuscript provides the description for a study completed assessing potential differences in estimated repetitions to failure when using BFR during inter-set rest or using traditional inter-set rest periods without BFR. The results support the hypothesis that there would be no difference between the types of rest intervention used.

1. Were participants instructed to maintain their same dietary intake before experimental session 2 and 3? This seems like a confounding variable that should be expanded on in the limitations section of the discussion. That is, could dietary intake have changed substantially between the two experimental sessions?

2. Was statistical normality determined for the data in this study?

3. I am confused as to why a Spearman's correlation (rho) method was used with these particular data sets. Is there a rationale for not using a Pearson correlation coefficient?

4. Table 1 - what does "Squat" refer to? Does this the number of times/week they performed squat exercises?

5. Table 2 - the Spearman correlations have *, **, ***; however they have no explanation as to what this means statistically in the table.

Author Response

We thank you for reviewing our manuscript and for the constructive comments that have enabled us to improve the manuscript.

  1. Were participants instructed to maintain their same dietary intake before experimental session 2 and 3? This seems like a confounding variable that should be expanded on in the limitations section of the discussion. That is, could dietary intake have changed substantially between the two experimental sessions?

Authors’ response: Thank you for this insightful comment. As mentioned in the manuscript, participants were instructed to abstain from consuming caffeine or pre-workout supplements within 2–3 hours before the session and avoid eating within 1 hour before the visit. The actual diet before the two experimental sessions was not controlled nor was advice provided for participants to follow the same diet. We agree that differences in diet before each experimental session may have influenced the study results and have included this point as a limitation in the Discussion section (see below).

“Finally, participants were not instructed to follow a specific diet before each experimental session which may have influenced their physical performance and perceptual responses. The potential confounding factors which have been acknowledged may assist future studies with enhancing the robustness and validity of their findings.”

  1. Was statistical normality determined for the data in this study?

Authors’ response: Data was normally distributed, and the following has been added to the manuscript.

“Data normality was assessed via the Shapiro–Wilk test, considering skewness and kurtosis measures within acceptable ranges (kurtosis < +2.0, > −2.0; skewness between +2.0 and −2.0) [16].”

  1. I am confused as to why a Spearman's correlation (rho) method was used with these particular data sets. Is there a rationale for not using a Pearson correlation coefficient?

Authors’ response: We have re-analyzed data using the Pearson correlation coefficient and made amendments to the manuscript.

  1. Table 1 - what does "Squat" refer to? Does this the number of times/week they performed squat exercises?

Authors’ response: Yes this refers to the number of times/week the squat exercise is performed. We have amended the manuscript to make this clear.  

  1. Table 2 - the Spearman correlations have *, **, ***; however they have no explanation as to what this means statistically in the table.

Authors’ response: Thank you for identifying this error. We have now explained these symbols.

We trust that the issues above have been addressed and clarified sufficiently.

Reviewer 2 Report

Comments and Suggestions for Authors

This paper examines the effects of performing BFR during breaks in training. I am convinced that the topic is very interesting and can be applied to the sports field. On the other hand, questions remain regarding the physiological significance of BFR intervention during breaks and the main outcome, accuracy of estimated repetitions to failure. Therefore, I would like you to review the following points.

Although there is a reference to BFRrest in the introduction section, I am aware that BFR is usually performed during training. Are there any differences in physiological or training effects related to this normal use of BFRrest? By mentioning this point, the reason why this study focused on BFRrest will become clearer.

In addition, this study focuses on the accuracy of estimated repetitions to failure. It is a well-known fact that training volume is important for training effectiveness. Therefore, it would have been easier to understand the significance of this study if we were comparing repetition numbers.

In the correlation analysis, I am not a statistician, so please forgive me if I am off the mark, but the number of plots in Figure 3 suggests that all the repeated data from the same subjects were used for the correlation analysis. Therefore, this analysis method includes multiple data from the same subject as well as data from different subjects. Is this a correct statistical treatment?

Author Response

We thank you for reviewing our manuscript and for the constructive comments that have enabled us to improve the manuscript.

  1. Although there is a reference to BFRrest in the introduction section, I am aware that BFR is usually performed during training. Are there any differences in physiological or training effects related to this normal use of BFRrest? By mentioning this point, the reason why this study focused on BFRrest will become clearer.

Authors' response: There is limited research on the effect of BFRrest, however, there is some evidence that this method of training intensifies metabolic and hinders recovery processes (which was emphasized in the first paragraph of the Introduction). However, we agree that greater emphasis is needed concerning the reason behind the study. The following has now been added to the Introduction.

“Since BFRrest is a novel BFR practice that may be used by resistance trainers to promote a greater exercise stimulus, it is of interest to investigate whether heightened physiological responses influence ERF accuracy and thus, impact monitoring of exercise performance.”

  1. In addition, this study focuses on the accuracy of estimated repetitions to failure. It is a well-known fact that training volume is important for training effectiveness. Therefore, it would have been easier to understand the significance of this study if we were comparing repetition numbers.

Authors’ response: We do agree with your point, however, will be unable to directly report this data in the manuscript. The reason that the repetition numbers are not reported in this manuscript is that this was the primary outcome of another manuscript that is currently under review in another journal. However, as emphasized throughout the manuscript, this paper is focused on the accuracy of using the ERF scale. We believe that not reporting the actual repetitions will not be a hindrance to the readers when interpreting the findings and understanding the significance of the study.

  1. In the correlation analysis, I am not a statistician, so please forgive me if I am off the mark, but the number of plots in Figure 3 suggests that all the repeated data from the same subjects were used for the correlation analysis. Therefore, this analysis method includes multiple data from the same subject as well as data from different subjects. Is this a correct statistical treatment?

Authors’ response: This is suitable given that all subjects completed the same number of sets and therefore each subject has the same number of data points. We have used this statistical method previously (see reference below).

Hackett DA, Cobley SP, Halaki M. Estimation of Repetitions to Failure for Monitoring Resistance Exercise Intensity: Building a Case for Application. J Strength Cond Res. 2018 May;32(5):1352-1359. doi: 10.1519/JSC.0000000000002419. PMID: 29337829.

We trust that the issues above have been addressed and clarified sufficiently.

Round 2

Reviewer 2 Report

Comments and Suggestions for Authors

Thank you very much for your response.

Congratulations!! I have no more comments for this paper.